# Effect of 12-Week Swimming Training on Body Composition in Young Women

**DOI:** 10.3390/ijerph16030346

**Published:** 2019-01-26

**Authors:** Malgorzata Charmas, Wilhelm Gromisz

**Affiliations:** Faculty of Physical Education and Sport in Biała Podlaska, Josef Pilsudski University of Physical Education in Warsaw, 2 Akademicka Str., 21-500 Biała Podlaska, Poland; wilhelm.gromisz@awf-bp.edu.pl

**Keywords:** swimming training, body composition, aerobic exercise

## Abstract

Background: Systematic physical activity can permanently prevent disadvantageous developments in the human body. This is very important especially for women, for whom the maintenance of a lean body in good shape is sometimes a primary consideration. However, in most cases, this activity is taken randomly and does not produce the desired effects such as reducing body fat. The purpose of the study was to evaluate changes in female body composition induced by 12 weeks of swimming training compared to sedentary controls. Methods: Training sessions occurred three times per week (60 min/session). Height, body mass, and waist/hip circumference and waist/hips ratio (WHR) were measured. Body cell mass (BCM), total body water (TBW), extracellular (ECW) and intracellular water (ICW), fat mass (FM), lean mass (FFM), and muscle mass (MM) were measured using bioelectrical impedance (pre/post). Results: Training elicited decreases in hip circumference and increase in WHR. No changes were recorded in BCM, TBW, ECW, ICW, FM, FFM, and MM. Controls experienced decreases in values of BCM, ICW, and MM and increases in ECW. Conclusion: The applied swimming training did not significantly affect the body composition parameters. Inactivity also triggered a tendency toward unhealthy movement of water from the intracellular to extracellular space.

## 1. Introduction

Recently, interest in the composition of the human body has been significantly growing with emphasis on maintaining a good ratio of components such as fat mass and lean body mass including muscle mass and water [1,2]. The study of body composition in conjunction with the analysis of endocrine have become an important element not only assess the risk of diseases associated with obesity, but also in their prevention and treatment (e.g., diet, exercise/training) [3]. In competitive sport the knowledge of body tissue proportions is essential to determine the morphological characteristics of an athlete. The opportunity for excellence in sport practice depends on this very information [3,4,5].

Low physical activity of humans leads to changes in the proportions of the basic components of the body [6,7]. It is well known that this refers mainly to the increase in body fat [8]. Numerous studies have shown that excess body fat and its specific depositing, depending on gender, significantly increases the risk of ischemic heart disease, diabetes, hypertension, or various forms of cancer [9,10,11,12,13,14]. The effects of an imbalance in the body of individual body components affects people of all ages. Preservation of the content of body components from an early age ensures the maintenance of good health later in life, when the metabolic processes are slowed down [15,16,17].

Moderate, but systematic physical activity can permanently prevent disadvantageous developments in the human body [18,19]. This is very important especially for women, for whom the maintenance of a lean body in good shape is sometimes a primary consideration. However, in most cases, this activity is taken randomly and does not produce the desired effects such as reducing body fat. Even the best knowledge of various techniques of exercises without any knowledge of changes occurring in the body under the influence of an exercise will not ensure attainment of goals. Therefore, the ability to monitor (interpret) changes occurring in body composition should be incorporated into the training process, not only for athletes but also for people undertaking physical exercise for health. Therefore, there is a need to analyze the effects of various types of physical activity especially one undertaken by women participating in various activities (aerobics, jogging, Nordic walking, swimming, etc.) on the components of their bodies. Results of such analyses would help to plan a training process with the possibility of modifying it at each stage, thus leading to attainment of the expected result.

The primary objective of this research was to assess the impact of a 12-week swimming training (involving three sessions per week) on the components women’s bodies aged 20–21 years. It was assumed that the primary objective of the experiment would be attained by answering the following questions: (1) Will the 12-week swimming training cause changes in body composition of the young, healthy women in comparison to the sedentary women? and (2) What is the direction and significance of changes in body composition caused by the 12-week swimming training in comparison to the sedentary life style? It is assumed that the applied training program will induce beneficial changes in body composition components in the direction of decreasing fat content while increasing lean body mass.

## 2. Material and Methods

### 2.1. Participants

Thirty-four young women took part in the experiment and they all consented to participation. The participant were students of Cosmetology and Tourism and Recreation courses (1st and 2nd year). The research was approved by the Ethics Committee of the University of Physical Education in Warsaw (SKE 001-37-1/2007).

Before the experiment, all subjects were asked to complete a questionnaire to assess their health and physical activity. Exclusion criteria included: (1) ingestion of medication (especially contraceptives and large amounts of vitamins), (2) ingestion of alcohol/ethanol (at least 2 times a day), and (3) cigarette smoking. Additionally, the subjects were asked to analyze their menstrual cycles by measuring vaginal temperature before getting out of bed in the morning [20]. Based on this data, healthy young women not actively involved in any sport and showing no abnormalities of the menstrual cycle were selected. Additionally, all participants attended the physical education classes within the curriculum provided for the courses (2 × 45 min per week) and entering the study had the ability to swim. The selected women were randomly divided into two groups: a swimming group (SG, *n* = 17, aged 21 ± 1 year) and a non-swimming group (NSG, *n* = 17, aged 20 ± 1 year). The anthropometric characteristics of both groups are shown in Table 1. In addition, all subjects were asked not to make any changes to their daily diet and not to engage in any dietary practices that would result in weight loss. A similar procedure was applied in studies by Mikkola et al. [21], Park et al. [22], and Sillanpaa et al. [23]. The participants did not know about the main purpose of the study. That is why it can be assumed they did not change their habits such as daily diet to help to improve the result.

### 2.2. Methods

The body height and waist and hip circumference of each subject were measured by generally accepted methods accurate to the nearest 0.1 cm. Body mass was determined using an electronic scale (Tanita BF-666, TANITA, Tokio, Japan, accuracy—0.05 kg). Body mass index (BMI) (kg/m^2^) was calculated from the formula BMI = m/h^2^, where m—body mass (kg), and h—body height (m). The components of the body: body cell mass (BCM, (kg)), total body water (TBW, (L)), extracellular (ECW, (L)) and intracellular water (ICW, (L)), fat mass (FM, (kg)), lean body mass (FFM (kg)), and muscle mass (MM, (kg)) were measured using the method of bioelectrical electrical impedance analysis (BIA) for body composition analysis with appropriate software (Akern BIA-101, Akern SRL, Florence, Italy). All the subjects were instructed as to the proper preparation prior to measurement, i.e., no alcohol for at least 48 hours before the test, emptying the bladder and not drinking any fluids for at least 2 hours before. Menstruation and taking of any medication disqualified participants from performing the test measurements on that day. All measurements were carried out after a 10-h overnight fast before the beginning of the experiment (1) and after (2). Over the data collection period outcome assessors were blinded to the participants group allocation. 

### 2.3. Training Protocol

The women from the swimming group (SG) participated in 12-weeks of swim training, (classes 3 times a week, 60 min./class.). The training was held the swimming pool of the Faculty of Physical Education and Sport in Biała Podlaska and was instructed by a qualified person according to a program prepared for untrained participants. During the experiment, constant temperature conditions were maintained in the swimming environment. Air and water temperature were 27.5 °C and 30.5 °C respectively. During the experiment, 36 training sessions were held during which the subjects swam a total of 42,000 m (±3000). The intensity of the training performed was at 60–80% of HR_max_. In all the training sessions the participants’ heart rates were monitored by a waterproof heart rate monitor (Polar Electro Oy, Kempele, Finland). A sample profile of heart rate of a selected subject and the structure of the training session are shown in Figure 1. To further assess the intensity of the exercise sessions, we assessed blood lactate concentration before and after a single session at both the beginning and end of the study.

### 2.4. Statistical Analysis

The results were statistically analyzed using the program *Statistica* for statistical computing, v. 6.0. For the analyzed parameters average values were calculated ($\bar{x}$) with standard deviations (SDs) and expressed in the form of $\bar{x}$ ± SD, giving additional value ranges. Statistically significant changes were assumed to be at the level of *p* < 0.05. The fulfilment of the principle of randomization was established, when qualifying for the experimental and control groups using one-way ANOVA analysis of variance, with which there was no statistically significant differences between parameters such as: body weight (F = 0.22, *p* < 0.05) and BMI (F = 0.22, *p* < 0.05). Any deficiencies in the results that emerged due to objective reasons (e.g., illness, menstruation) were supplemented by using the method of Ferguson-Takane [24]. After checking the coefficient of variation (%CV) for the studied parameters, log10 data transformation was performed to normalize their distribution. One-way ANOVA analysis of variance was performed within groups in order to examine the significance of differences between the averages of the data collected on the two test days (pre- and post-training). In order to determine the effect of the 12-week training program, *t*-tests were used to compare the mean values of the parameters. In order to analyze the interaction between the study parameters, a Pearson’s linear correlation analysis was performed.

## 3. Results

Table 1 shows the anthropometric characteristics of the women from both groups. The body composition parameters of both groups are presented in Table 2. All variables are presented as mean values (±SD) together with their range (min/max) and the calculated pre-post change (Δ, Δ%). It is noteworthy that at the beginning of the experiment the SG group had 24 women, but the principle was adopted that eventually would form the SG group of these women (*n* = 17) who participated in at least 95% of all 36 training sessions (seven women resigned due to health problems or personal reasons).

### 3.1. Anthropometric Characteristics

The 12-week swimming training elicited a statistically insignificant decrease in body mass of the SG subjects (average: 0.14 ± 1.83 kg) (Table 1). That result was supported by a modest decrease in the BMI parameter (average: 0.1 ± 2.9%). As a result of the training, the average hip circumference of the SG subjects significantly decreased, as well (average: 1.6 ± 1.4% *p* = 0.0038). In the case of waist circumference, a negligible increase was recorded (average: 0.02 ± 3.7%). The direction and significance of changes in the circumferences of the hips and waist were reflected in the values of the waist/hip ratio (WHR). This parameter significantly increased within this group (average: 1.84 ± 2.68%, *p* = 0.02).

Within the NSG group, an insignificant increase in body mass was observed (average: 0.31 ± 1.08 kg), which was reflected in the direction of changes in the BMI parameter (average insignificant increase of 0.5 ± 1.8%). There was also a statistically insignificant increase in circumferences of waist and hips (0.5 ± 4.1%, 0.8 ± 3.3%, respectively).

### 3.2. Body Composition Measurements

The 12-week swim training protocol elicited a decrease in water content in the tested compartments of the SG group. There was an insignificant decrease the total body water (TBW) by an average of 0.5 ± 1.7%, extracellular water (ECW) by an average of 1.8 ± 4.1% and intracellular water (ICW), by an average of 0.3 ± 2.9%. The applied training contributed to a significant decrease in body fat (FM) in the SG group by an average of 0.2 ± 9.5%. At the same time there were an insignificant increase in fat-free body mass (FFM) by an average of 0.2 ± 2.7% and muscle mass (MM) (average: 0.2 ± 2.1%). The applied training resulted in a significant decrease in body cell mass (BCM), by an average of 0.2 ± 2.7%.

In the case of the NSG group, 12 weeks of physical inactivity caused an insignificant reduction in TBW by an average of 0.5 ± 2.2% while the volumes of ECW and ICW were statistically significantly reduced by an average of 3.3 ± 4.1% (*p* = 0.018) and 3.1 ± 2.7% (*p* = 0.0033), respectively. There was an insignificant increase in FM by an average of 1.4 ± 3.5%. A statistically insignificant decrease in FFM by an average of 0.4 ± 2.0% was also observed. Muscle mass was significantly decreased by an average of 2.5 ± 2.4% (*p* = 0.0062). In addition, there was a significant decrease in body cell mass (BCM), by an average of 2.9 ± 2.5% (*p* = 0.0029).

In the swimming group (SG), analysis of the Pearson linear correlation showed a directly proportional correlation (*p* < 0.05) between the change in body mass and the changes in waist circumference (*r* = 0.61), hip circumference (*r* = 0.62), FM (*r* = 0.92), and TBW (*r* = 0.62). Additionally, it was found that changes in waist and hip circumference correlated with changes in body mass index BMI (*r* = 0.61, *r* = 0.64, respectively). It was demonstrated that changes in BCM correlated highly with the changes in MM (*r* = 0.86), which in turn correlated highly with the changes in ICW (*r* = 0.86). On the other hand, changes in FFM were influenced by the changes in TBW (*r* = 0.76), which correlated with the changes in ECW (*r* = 0.65).

Within the NSG group, analysis of the linear Pearson correlation showed indirectly proportional correlation (*p* < 0.05) between the change in body mass and changes in waist circumference (*r* = 0.64) and hip circumference (*r* = 0.80). Additionally, it was found that both changes in the waist and hip circumference correlated with changes in body mass index BMI (*r* = 0.63, *r* = 0.81, respectively). Changes in BCM were correlated with changes in TBW (*r* = 0.69) and FFM (*r* = 0.71), and remained, as expected, highly correlated with the changes in ICW (*r* = 0.99) and MM (*r* = 0.99). In addition, there was correlation between the changes in TBW and the changes in ECW (*r* = 0.69) and ICW (*r* = 0.69). It was also demonstrated that the change in the volume of ICW depended on the changes in FFM (*r* = 0.71), which in turn was correlated with the changes in MM (*r* = 0.79).

Analysis of significance of the changes in mean values of the studied anthropometric parameters and body composition between groups: SG and NSG showed that the 12-week swimming training elicited significant changes (*p* < 0.05) in the following parameters: waist circumference, body cell mass (BCM), extracellular (ECW), and intracellular (ICW) water and muscle mass (MM).

## 4. Discussion

The purpose of study was to evaluate changes in the body composition of young females who underwent 12-weeks of swim training.

It is well known that prolonged exercise training of moderate intensity is the best way not only to maintain a healthy body weight, but also improve general health [25]. In the present study, the evaluation of heart rate (HR) during all training sessions allowed the confirmation of submaximal exercise intensity. During the training sessions, heart rate did not reach maximum values and on average was maintained at 60–80% of the maximum heart rate, which roughly corresponded to an oxygen consumption of 55–65% VO_2_max. Furthermore, although the applied effort during the training sessions induced an increase in blood lactate concentration (LA) in most tested women (90%), this parameter never exceeded the anaerobic threshold at either test point 1 or 2 (4 mmol/l). One can be reasonably assured that the energy used during the exercise sessions came from aerobic processes. Furthermore, it was noted that change in the concentration of LA induced by a single swimming session during the session at the end of experiment (12th week) was significantly lower (*p* < 0.05) than at the beginning of the experiment. This is likely a sign of adaptation towards more efficient use of the oxygen supplied to the muscles as the threshold of anaerobic changes shifted toward higher loading values.

The training protocol also resulted in a change in the deposition of fat in the SG group participants with a significant decrease in hips circumference (*p* < 0.01) and increase in waist/hip ratio (WHR) (*p* < 0.05) (waist circumference relatively unchanged). This trend shows the beneficial effect of this type of activity on body circumferences. Considering the decrease in body mass and the direction of changes of these anthropometric characteristics, it seems that this type of physical activity likely inhibits the deposition of fat in the upper parts of the body. Lower body fat deposition in this area can lead to a lowered risk of disease particularly cardiovascular disease [9,10]. This was confirmed in the behavior of the WHR coefficient (Table 1). A similar dependence was observed by Park et al. in their studies [22]. No changes in these parameters were recorded within in CG group. These findings seem to be in accordance with previous findings but in older adults [26,27].

In the case of the women from the NSG group, an insignificant increase in waist, and hip circumference as well as body weight was observed. Although these changes are not statistically significant, maintenance of such a trend could, over the course of time, lead to a significant increase in body weight, which, as already mentioned above, is associated with high risk of many chronic diseases. Such a trend can be observed by analyzing the direction of changes in the body mass index BMI.

It is well known that the accumulation of fat (FM) in relation to fat-free body mass (FFM) is important especially for women because of the need for hormonal homeostasis, which in turn ensures the correct operation of the menstrual cycle [3,28,29]. Appropriate body fat content helps to preserve the overall balance of the body through the production of many biologically active molecules called adipokines that ensure the proper functioning of the immune system [30], the endocrine system [31], and many metabolic processes [32]. One of the substances produced by adipose tissue is leptin, a protein resource providing communication between the fat and the appetite center in the hypothalamus, which in turn provides effective control of appetite [33]. The content of fat tissue in the SG women deviated from the generally accepted healthy standards (16–20%) in the direction of increased adiposity (mean 26.3 ± 4.7%) at the expense of fat-free body mass (average 73.7 ± 4.6%, healthy standard: 78–80%). A similar trend was observed in the women from the control group (NSG). The slight decrease in body mass of the SG women was likely the result of a slight reduction in FM and TBW content while increasing the content of FFM including MM. A similar trend was observed by Kersick et al. [34] who subjected healthy obese women to a 14-week strength training program. On the other hand, Martins et al. [35] did not observe any changes in body composition of healthy women (30 ± 12 years, BMI = 22.7 ± 2.3 kg/m^2^) participating in a 6-week moderate training program (4 times a week, 65–75% HR_max_). An interesting experiment was conducted by Arroyo-Toledo et al. where 20 trained girls (16.1 ± 1 year, BMI = 25.1 ± 1.8 kg/m^2^) took part in 10-week swimming training and were divided into two groups traditional periodization (BP) and reverse periodization (RP) [36]. The authors recorded a significant increase in FFM (*p* < 0.05) and decrease in FM (*p* < 0.05) within BP group while they did not notice any changes within RP group [36]. Siqueira et al. did not observe any changes in body composition of women (*n* = 133) with rheumatoid arthritis participating in a 16-week controlled program consisting of different kind of aquatic exercise (3 sessions per week) [37]. Although, Penaforte et al. noticed that short-term water aerobics program (8 weeks, 3 sessions per week) in obese women (BMI = 36.1 ± 6.3 kg/m^2^) caused a significant reduction of weight, BMI, fat mass, arm circumference, and hip circumference [38]. Rica et al. carried out a program which consisted of water-based exercise (12 weeks, 3 sessions per week) in obese older women (65–75 years, BMI > 30 kg/m^2^). They noticed no changes in body composition parameters but an improvement in all functional parameters (i.e., time to walk, chair test before and arm flexion, and self-reported quality of life) [39]. In the other study, Pereira et al. conducted a 12-week water aerobics program in older adults (58.80 ± 14.32 years, 75.54 ± 15.53 kg of body mass) [26]. They recorded reduce body mass, loss of fat mass. In the NSG group, all women experienced an insignificant increase in body weight, which was likely associated with an insignificant increase in FM and a significant (*p* < 0.05) increase in TBW. There was an insignificant decrease in FFM and statistically significant decrease in MM (*p* < 0.01). Maintaining such a trend could lead to an increased risk of metabolic disorders, as noted above.

In the present study we analyzed the parameter, body cell mass (BCM) which recently has been successfully used to assess the nutritional status of cells. This in turn allows the estimation of the state of MM and tissue proteins [40]. BCM is a component of body composition, which is responsible for the consumption of oxygen and carbon dioxide production. It represents the total mass of living, metabolically active cells. A decrease of BCM has been observed in patients with sarcopenia or anorexia [41]. A healthy level of BCM for age of women participating in this study (<30 years) is >49% [42]. The BCM content in all studied women averaged 48.4 ± 1.8%. After 12 weeks of the experiment a slight, non-significant decrease in the nutritional status of cells in the SG group was observed and a statistically significant decrease (*p* < 0.01) was seen in the NSG group (NSG).

It is well known that water plays a versatile role in the living organism. It is a means of intra organism transport of: nutrients, hormones, enzymes, and others. It participates in maintaining a constant body temperature and blood pressure regulation. The total body water (TBW) remains constant, even for many years. Maintaining an appropriate amount of TBW with reference to volume of ECW to ICW ratio ensures the maintenance of normal body function [43]. In the case of the women participating in this experiment we found TBW values lower than the norm (normal: 62%) (participant values: 55.5 ± 2.9%). It was noted that the 12 weeks of swimming training elicited a slight decrease in TBW for all the exercising women, with a concurrent slight increase of its percentage (average: 0.18 ± 1.6%). Similar results were obtained by Quiterio et al. [44] for young female athletes (13 years) undergoing training of varying duration (<4.5h/week, 4.5–8.9 h/week and 9 h/week). The 12-week swimming training caused a negligible decrease of both ECW and ICW. The decrease in ICW volume in the SG group, despite a slight increase in MM was likely due to, probably, insufficient water supply.

In the NSG group, the 12 weeks of inactivity caused a similar slight decrease in TBW content, as in the case of the SG group. It is worth emphasizing that this lack of activity caused a further significant decrease in the volume of ICW (*p* < 0.01), with a concurrent significant increase in the volume of ECW (*p* < 0.05). It can be assumed that the disturbance of water homeostasis in the bodies of the NSG women, due to the reduction in TBW and the shifting of water from the intracellular to the extracellular compartment, will continue to deepen due to the lack of sufficient physical activity and is likely to pose a risk of dangerous to their health in the future (swelling) [45].

The subject of changes in body composition components under physical activity is widely undertaken by many authors, which is untestable because of its beneficial effects on the widely understood human health. The parameters of these components are modulated by various factors, which are usually the type of training (type of exercise, intensity, duration), diet (exclusion or introduction of new foods). In addition, the test criteria and methods and the condition of their conduct have a decisive influence on the results. In the case of this study, participants were young, healthy women who had a normal body weight. Furthermore, the lack of daily diet analysis prevented an objective assessment of the results obtained. The obtained results did not show the effect of the 12-week swimming training on body composition components while significantly improving the performance of the organism (unpublished data). Nevertheless, the authors are sure that the data obtained from the applied training program provide a strong basis for conducting the same experiment in women with varying degrees of body fat (overweight, obesity) in controlled conditions of the daily diet. This could, in turn, be a starting point for programming physical efforts to achieve the objectives: improving physical fitness, reducing body fat and thereby reducing metabolic disorders caused by excess.

## 5. Limitations

In the presented study we attempted to assess the impact of a 12-week swimming training program on the body composition of young women. We realize that a limitation of this study is that laboratory conditions were not fully observed (lack of energy analysis of meals). We merely had obtained assurance of the subjects that they would not implement any changes in their diet, especially with respect to calorie restriction. In addition, the measurements were performed without regard to menstrual cycle day of each individual subject. Only the women who did not show any abnormalities of the menstrual cycle were qualified to have their measurements taken.

It is undeniable that for exercise to bring expected results, it should be supported by an appropriately selected diet, especially bearing in mind age and the health condition. Control of the effectiveness of training and its impact on the body composition of the participants of such training in conjunction with a healthy diet will be the subject of our research in the future.

## 6. Conclusions

Prolonged aerobic training (12 weeks of swimming) caused no significant change (decrease) in body weight of the tested women. The applied training resulted in an insignificant decrease in body fat tissue, with a concurrent an insignificant increase in lean body mass, (including muscle mass). In the case of the SG group, body shape improvement was noted in the form of reduction in hip circumference.

The beneficial effects of physical activity become especially apparent in the case of the NSG group, where the lack of this activity caused weight gain and an increase in body fat content with a concurrent decrease in lean body mass (including muscle mass). In addition, the 12-week physical inactivity showed a tendency toward unhealthy movement of water from intracellular to extracellular space. The lack of physical activity resulted in increased waist and hip circumference, which indicates an increased deposit of visceral fat tissue.

Additionally, accidentally taken, especially by women, physical effort in most cases does not bring the expected results (no reduction of body fat). Therefore, great importance is attached to training the appropriate team of people who, thanks to their knowledge and skills, can provide their wards with conditions to meet all expectations. Each study supplementing the knowledge of coaches/instructors allows them to optimize the training process in such a way that the persons undertaking the selected physical activity have the certainty of achieving the intended goal, in this case shaping the figure.

## Figures and Tables

**Figure 1 ijerph-16-00346-f001:**
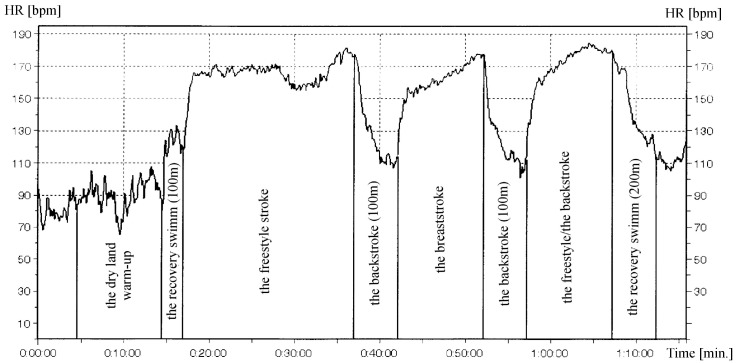
One of the heart rate (HR) profiles and structure of one of the single sessions during 12-week swimming training.

**Table 1 ijerph-16-00346-t001:** Characteristic of the examined groups: SG (swimming group) and NSG (non-swimming group). Rest values pre (1) and post (2) 12-week swimming training. The values are given as mean ± SD. Δ%—percentage change.

Parameter		SG				NSG			
		1	2	Δ	Δ [%]	1	2	Δ	Δ [%]
**Age** (years)	$\bar{x}$ ± SD	21 ± 1				20 ± 1			
	range	19–23				19–22			
**height** (cm)	$\bar{x}$ ± SD	169.7 ± 4.8				164.9 ± 4.2			
	range	158.5–176.5				160.0–173.0			
**body mass** (kg)	$\bar{x}$ ± SD	63.1 ± 7.2	63.0 ± 7.0	−0.14 ± 1.83	−0.15 ± 2.92	62.3 ± 8.0	62.6 ± 7.9	0.31 ± 1.08	0.53 ± 1.8
	range	51.7–78.6	52.9–79.4	−3.00–2.80	−4.7–4.6	46.5–78.6	46.3–79.0	−1.40–2.00	−2.1–3.6
**waist** (cm)	$\bar{x}$ ± SD	70.8 ± 5.2	70.9 ± 5.2	0.09 ± 2.58	0.2 ± 3.7	73.5 ± 5.4	73.8 ± 5.8	0.37 ± 2.87	0.5 ± 4.1
	range	65.0–82.0	64.5–84.0	−4.00–5.00	−5.6–7.7	62.0–81.0	60.0–83.5	−2.50–6.80	−3.3–10.1
**hips** (cm)	$\bar{x}$ ± SD	97.7 ± 5.3	96.1 ± 5.4	−1.59 ± 1.41	−1.6 ± 1.4 **	98.8 ± 7.0	99.6 ± 6.7	0.75 ± 3.13	0.80 ± 3.3
	range	92.5–112.0	92.0–11.5	−4.0–0.0	−4.1–0.0	87.0–110.0	85.0–112.0	−3.00–7.60	−2.8–8.3
**WHR**	$\bar{x}$ ± SD	0.72 ± 0.03	0.74 ± 0.02	0.01 ± 0.02	1.84 ± 2.68 *	0.74 ± 0.04	0.74 ± 0.02	0.0 ± 0.03	−0.23 ± 4.18
	range	0.68–0.77	0.70–0.77	−0.01–0.05	−7.76–7.69	0.70–0.81	0.70–0.79	−0.05–0.05	−6.62–6.36
**BMI** (kg·m^-2^)	$\bar{x}$ ± SD	21.8 ± 2.0	21.4 ± 1.9	−0.04 ± 0.63	−0.1 ± 2.9	22.9 ± 3.1	23.0 ± 3.1	0.12 ± 0.4	0.5 ± 1.8
	range	18.9–27.0	19.0–27.3	−1.05–0.92	−4.7–4.6	18.2–29.7	18.1–30.1	−0.5–0.74	−2.1–3.6

WHR—waist–hips ratio; BMI—body mass index; * change is significant *p* < 0.05; ** change is significant *p* < 0.01.

**Table 2 ijerph-16-00346-t002:** Rest values of body compositions parameters in the examined groups: SG (swimming group) and NSG (non-swimming group); rest values pre (1) and post (2) 12-week swimming training. The values are given as mean ± SD, Δ%—percentage change.

Parameter		SG				NSG			
		1	2	Δ	Δ [%]	1	2	Δ	Δ [%]
**BCM** (kg)	$\bar{x}$ ± SD	22.5 ± 1.6	22.4 ± 1.7	−0.04 ± 0.61	−0.2 ± 2.7	21.4 ± 1.5	20.8 ± 1.4	−0.63 ± 0.53	−2.9 ± 2.5 **
	range	19.1–24.5	18.4–24.1	−1.10–0.90	−4.5–3.9	18.8–23.8	18.3–23.5	−1.30–0.60	−5.9–3.0
**TBW** (L)	$\bar{x}$ ± SD	34.9 ± 2.5	34.7 ± 2.5	−0.19 ± 0.61	−0.5 ± 1.7	33.3 ± 2.7	33.1 ± 2.9	−0.16 ± 0.73	−0.5± 2.2
	range	29.7–38.6	29.9–38.2	−0.90–1.20	−2.6–3.3	29.1–38.4	28.7–38.6	−1.30–1.30	−3.7–3.9
**ECW** (L)	$\bar{x}$ ± SD	14.6 ± 1.3	14.3 ± 1.2	−0.28 ± 0.58	−1.8 ± 4.1	13.7 ± 1.6	14.2 ± 1.8	0.45 ± 0.53	3.3 ± 4.1 *
	range	12.3–16.9	13.1–16.6	−0.90–0.80	−6.2–6.5	10.4–16.7	11.3–17.6	−0.50–1.20	−4.0–8.9
**ICW** (L)	$\bar{x}$ ± SD	20.5 ± 1.5	20.4 ± 1.6	−0.05 ± 0.59	−0.3 ± 2.9	19.6 ± 1.4	19.0 ± 1.3	−0.61 ± 0.53	−3.1 ± 2.7 **
	range	17.4–22.4	16.8–22.0	−1.10–0.90	−1.10–4.3	17.1–21.7	16.6–21.5	−1.30–0.60	−6.5–3.2
**FM** (kg)	$\bar{x}$ ± SD	17.1 ± 5.1	16.9 ± 5.3	−0.12 ± 1.62	−0.2 ± 9.5	17.9 ± 6.4	18.2 ± 6.6	0.31 ± 0.46	1.4 ± 3.5
	range	11.7–29.3	11.6–31.1	−2.70–1.80	−13.5÷ 12.1	6.8–29.5	6.9–30.2	−0.70–1.10	−7.1–7.8
**FFM** (kg)	$\bar{x}$ ± SD	46.1 ± 2.9	46.2 ± 2.9	0.10 ± 0.75	0.2 ± 1.6	43.9 ± 2.8	43.7 ± 3.0	−0.16 ± 0.87	−0.4± 2.0
	range	39.8–50.4	39.9–50.0	−1.0–1.30	−2.0–2.7	39.7–48.5	39.3–48.8	−1.50–1.60	−3.3–3.7
**MM** (kg)	$\bar{x}$ ± SD	27.6 ± 1.9	27.8 ± 2.1	0.06 ± 0.58	0.2 ± 2.1	26.4 ± 1.9	25.8 ± 1.9	−0.66 ± 0.64	−2.5 ± 2.4 **
	range	23.6–30.0	22.8–29.5	−0.80–1.10	−3.4–3.9	22.6–29.5	22.6–29.0	−1.50–0.90	−5.5–3.6

** change is significant, *p* < 0.01; * change is significant, *p* < 0.05. BCM—body cell mass; TBW—total body water; ECW—extracellular water; ICW—intracellular water; FM—fat mass; FFM—free fat mass; MM—muscle mass.

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
