# Peer review of "Effect of 12-Week Swimming Training on Body Composition in Young Women"

_ijerph, 2019, doi:10.3390/ijerph16030346_

Round 1
Reviewer 1 Report
The authors aimed to evaluate the body composition changes caused by 12 weeks of swimming, using some anthropometric measurements and body bioimpedance. This is a “clean”, easy to read and understand paper, with a simple design that accomplish the main aim of the study. So, congratulations for the authors. However, some rationale is missing about the relevance of the study. To my knowledge, this is nothing quite new, but still interesting. The authors can and should improve the introduction so that the readers can really understand the need for this investigation. Some interesting results were found but some doubts emerged when reading, about methodological issues that could compromise the outcomes, discussion and conclusion. Please see below.
Specific comments:
English writing should be improved and sometimes authors should be more specific. For example, what do authors mean as “unfavorable developments”?
The authors did not measure body “structure”, but only body composition using different methods. So, this should be changed in the manuscript.
In the methods, what do you mean as “large amounts of vitamins”?
Did the authors control ingestion? How it was controlled? This is a major issue of the study and we think that probably contaminated the results.
Moreover, the authors did not report if the participants knew about the main purpose of the study and if they knew what was the group they belonged to (Swimming or non-swimming). I am afraid that if they know procedures and objective, they will be influenced, even though unconsciously, and they will change some habits, such as eating, to help to improve the results. Please clarify this.
What was the swimming experience of the swimming group? Did they perform more or less training sessions than before? The non-swimming group used to practice swimming but stopped with the beginning of the research or did not used to do any physical activity? This is another major issue that could influence the results. Starting from zero is different of being experienced to change bodycomposition, as well as stopping from usual practice could influence the results. The authors need to clarify and discuss this in discussion section so that we can clearly understand your outcomes and conclude relevant scientific evidences.
The results in pre-test are different between groups? The authors should compare and report this.
Also, in the results, I was a little confuse about the significance reported in the Tables. The authors reported differences in the percentage differences. This was between groups? And, the percentage was different but not the absolute values? Did the authors compared the pre and post test values? These should be reported. Results section should be improved regarding this.
The authors should change “rest values” to “pre-training” or anything similar.
Only training session was evaluated and then compared. This is a limitation of the study. How it was relevant for interpreting the results? All the training sessions were similar? All the responses were similar? Please, clarify.
Please change “lactic acid” to blood lactate concentration.
The authors reported 60÷80% Hrmax. Please, confirm this is correctly reported.
Discussion and conclusions should be improved regarding the limitations previously reported, since it would allow misleading interpretation of the results.
Author Response
Reviewer 1 – answers
Thank you very much for so thorough review of our article. Thank you very much for all the comments that are very valuable to us and have been very helpful in improving our work.
Answers:
English writing should be improved and sometimes authors should be more specific. For example, what do authors mean as “unfavorable developments”?
Answer: English language was checked by my colleague (native speaker) from one of the universities in USA. Phrase “unfavorable developments” means negative/disadvantageous developments. I realize it is not so clear, so I changed it.
The authors did not measure body “structure”, but only body composition using different methods. So, this should be changed in the manuscript.
Answer: Expression „structure” was used in meaning „shape” because waist and hips circumference were analyzed but I agree with that it can be little confusing and that is why I changed it in the entire body text.
In the methods, what do you mean as “large amounts of vitamins”?
Answer: Expression “large amounts of vitamins” means doses exceeding the daily requirement (vitamines administration on your own etc.), for example in the case of vitamin C, doses exceeding 1000 mg may cause acidification of the body, which in turn leads to the movement of fluids between the compartments (ECW, ICW).
Did the authors control ingestion? How it was controlled? This is a major issue of the study and we think that probably contaminated the results.
Answer: We did not control ingestion. That is why we described this aspect of the study as one of the limitations and placed proper sentences in section “4. Discussion”.
Moreover, the authors did not report if the participants knew about the main purpose of the study and if they knew what the group was they belonged to (Swimming or non-swimming). I am afraid that if they know procedures and objective, they will be influenced, even though unconsciously, and they will change some habits, such as eating, to help to improve the results. Please clarify this.
Answer: thank you very much for that comment. It is very useful, and I agree with it. I placed proper sentence at the end of the section “2.1. Participants”.
What was the swimming experience of the swimming group? Did they perform more or less training sessions than before? The non-swimming group used to practice swimming but stopped with the beginning of the research or did not used to do any physical activity? This is another major issue that could influence the results. Starting from zero is different of being experienced to change bodycomposition, as well as stopping from usual practice could influence the results. The authors need to clarify and discuss this in discussion section so that we can clearly understand your outcomes and conclude relevant scientific evidences.
Answer: All participants attending the study had the ability to swim but they were not practise any sport discipline especially swimming. Information regarding physical activity of all participants is included in text (section “2.1. Participants”: “…healthy young women not actively involved in any sport and showing no abnormalities of the menstrual cycle were selected…” and of course all of them didn’t know about the main purpose of the study. I placed that information in section “2.1. Participants”.
The results in pre-test are different between groups? The authors should compare and report this.
Answer: The differences between groups were analysed. Results were described in section “3.2. Body composition measurements”.
Also, in the results, I was a little confuse about the significance reported in the Tables. The authors reported differences in the percentage differences. This was between groups? And, the percentage was different but not the absolute values? Did the authors compared the pre and post test values? These should be reported. Results section should be improved regarding this.
Answer: In Tables we showed differences as D for the absolute values and D% for the percentage values but within the group and they were described in section “3. Results”. I checked every single data in the entire section “3. Results” and improved all incompatibilities according to data in Tables.
The authors should change “rest values” to “pre-training” or anything similar.
Answer: Writing “rest values” I would like to emphasize that all data were collected in rest status, but I change that expressions in the body text.
Only training session was evaluated and then compared. This is a limitation of the study. How it was relevant for interpreting the results? All the training sessions were similar? All the responses were similar? Please, clarify.
The obtained data was analyzed as the effect of 12-week swimming training. An example of one training session is described in the text. All sessions were conducted in such a way as to maintain the appropriate level of loading (by using appropriate physical activities - exercises in the water and different styles of swimming). Please, forgive me, but we think it is not a limitation of our study.
Please change “lactic acid” to blood lactate concentration.
Answer: It was changed in the body text.
The authors reported 60÷80% Hrmax. Please, confirm this is correctly reported.
Answer: Yes, it is correct.
Discussion and conclusions should be improved regarding the limitations previously reported, since it would allow misleading interpretation of the results.
Answer: It was done.

Reviewer 2 Report
The paper “Effect of 12-Week Swimming Training on Body Structure and Composition in Young Women” aims to evaluate changes in female body structure/composition induced by 12 weeks of swimming training compared to sedentary controls. Main data suggests that training effects were positive in the exercisers, while controls gained fat mass and lost lean mass (including muscle mass). Inactivity also triggered a tendency toward unhealthy movement of water from the intracellular to extracellular space.
The paper is within the scope of the journal.
However, I have some doubts regarding:
- the rational of the study. Please rephrase the introduction section to fit the scope of the paper and highlight the main aim of the study.
- Please include new evidences regarding exercise and health in different populations and in different context as it has been described an important contribution of this type of exercise on health parameters (for instance, refer to:
. Rica et al. Effects of water‐based exercise in obese older women: Impact of short‐term follow‐up study on anthropometric, functional fitness and quality of life parameters. Geriatr Gerontol Int. 2012; 13(1): 209-214.
. Neiva, et al. (2018). The effect of 12 weeks of water-aerobics on health status and physical fitness: An ecological approach. PLoS ONE 13(5), e0198319.
- Please include more data regarding the background of the subjects regarding physical activity.
- Please include the hypothesis of the study.
- Please include some more practical references in the conclusions, for coaches and instructors.
Author Response
Reviewer 2 – answers
Thank you very much for so thorough review of our article. Thank you very much for all the comments that are very valuable to us and have been very helpful in improving our work.
Answers:
- the rational of the study. Please rephrase the introduction section to fit the scope of the paper and highlight the main aim of the study.
Answer: Introduction has been completed. The main aim study is included in the section “Introduction”.
- Please include new evidences regarding exercise and health in different populations and in different context as it has been described an important contribution of this type of exercise on health parameters (for instance, refer to:
. Rica et al. Effects of water‐based exercise in obese older women: Impact of short‐term follow‐up study on anthropometric, functional fitness and quality of life parameters. Geriatr Gerontol Int. 2012; 13(1): 209-214.
. Neiva, et al. (2018). The effect of 12 weeks of water-aerobics on health status and physical fitness: An ecological approach. PLoS ONE 13(5), e0198319.
Answer: I know the experiments which were conducted by Rica et al. and Neiva et al. but they examined obese older women aged between 60 and 75 years and 58.80 ± 14.32 years old, respectively and that is why we decided not to mention it because my participants were young women 21 ± 1 years. I respect your recommendation and placed them.
Thank you very much for that comment. I placed additional data obtained from other scientists.
- Please include more data regarding the background of the subjects regarding physical activity.
Answer: I placed more data regarding the background of the subjects regarding physical activity
- Please include the hypothesis of the study.
Answer: It has been done. The “Introduction” section.
- Please include some more practical references in the conclusions, for coaches and instructors.
Answer: It has been done.

Reviewer 3 Report
Comments by topic
Abstract:
Statement: “…This is very important especially for women, for whom the maintenance of lean body in good shape is sometimes a primary consideration…”
“…However, in most cases, this activity is taken randomly and do not produce the desired effects such as reducing body fat…”
Comment: Personal not referenced information
Results from Abstract doesn’t match to the displayed in the results section.
Introduction:
Line 34: “…Low physical activity of humans leads to changes in the proportions of the basic components of 35 the body. It is well known that this refers mainly to increase in body fat…”
Suggestion: Clarify the phase and insert the proper reference to that information
Line 35: “…Numerous studies have 36 shown that excess body fat and its specific depositing, depending on gender, significantly increases the risk of ischemic heart disease, diabetes, hypertension, or various forms of cancer [6]…”
Suggestion: Don’t use a hole book to reference information so specific like body fat accumulation (obesity) and its risk of certain diseases or cancer. I’d suggest looking for WHO, or AHA, FDA, a most proper source of information.
Line 37-40: “…The effects of imbalance in the body of individual body components affect people of all ages. Preservation of the content of body components from an early age ensures the maintenance of good health later in life, when the metabolic processes are slowed down..”
Suggestion: Reference that information.
Line 41-52: broad content not referenced
Suggestion: Do a proper referenced introduction, avoiding hole book, using chapters or original scientific papers, research data, health organs, systematic reviews, etc.
Line 50: “…by women participating in various activities...”
Comment: The statement gives the author an impression that the importance of the work it’s not related only to the human health or enhance the global health, but the women health. However, If the authors were actual focusing on this, thus I suggest introducing the manuscript pointing out female aspects that justify this approach (e.g. swimming) in women system and not obesity or because of the fat tissue accumulation. I believe that the fact of a female individuals has been elected for the study not necessarily indicates the hole work has to be constructed on the gender, it’s a sample characteristic.
Material and methods:
Line 56: (questions of the study)
“… 1) Will the 12-week swimming training cause changes in body composition of the tested women? and 2) What is the direction and significance of changes caused by the 12-week swimming training?...“
Suggestion: Correct that information to the evaluated, e.g. the primary object is in comparison to sedentary women, and so on. Because, starting from that informed, the control group would be no longer necessary, only a pre-post evaluation.
Line 66: “…Additionally, the subjects were asked to analyze their menstrual cycles by measuring vaginal temperature before getting out of bed in the morning…”
Comment: Why? Authors must clarify the reasons and give proper reference to that procedure.
Line 75: Table 1:
Ø OBS1: on title, I suggest change “…rest value (1) and after the experiment (2)…” to “rest values pre (1) and post (2) 12-week swimming training”
Ø OBS2: BMI SG: 21.9+2.2 is the same value to both moments and there was a decrease of 0.1%, I think that might be some mistake.
Line 91: Before and after all 12 weeks experiments or after every experiment? I suggest make this clear. I believe these data were collect twice, one on the exact beginning of the protocol and then 12-weeks later for the same investigator.
Line 100: about the intensity of the training: it was measured from Karvonen method? Taken everyday or based on the data of the initial test?
Figure 2: unnecessary.
Results:
Line 128: “…Table 1 shows the anthropometric characteristics of the women from both groups..”
Comment: phase can be removed or place the table 1 on results section.
Line 131-133: “…It is noteworthy that at the beginning of the experiment the SG group had more women, but the principle was adopted that eventually would form the SG group of these women (n=17) who participated in at least 95% of all 36 training sessions...”
Comment: The phase is confusing. What do you mean with “more women”? Isn’t 17 per group since the very beginning? Statement should be corrected with the relevant information.
Line 133: “…at least 95% of all 36 training sessions…”
Comment: what was the participation rate of the swimming group?
Line 166-182: (correlations analysis)
Comment: What was the moment evaluated? Final experiment moment? Correlations doesn’t show causality. I think the Persons can give additional information but, in this case, e.g. the BMI and hip and waist circumference, the BMI implies the mass (weight) evaluation and certainly a higher hip and waist circumference is positively correlated to BMI, there is some unnecessary correlations not related to the given approach/training i.e. showing correlation between swan kilometer per lost of WHR or to the enhancement of heart rate response. I would suggest a different Persons evaluation.
Line 183-186: Comparison between groups
Comment: There was no intervention on the control group, therefore, there is no reasoning question to compare CT to swimming. If the study was based on two different physical approaches (e.g. on water and off water, cooled vs heated swimming pool) so that would be a reason for this, but in this case there was not. The CT group in this study shows how 12-weeks of sedentary life-style can bring negative significant changes to the women’ body composition.
Results displayed on manuscript and the mistakes in comparison to the abstract
All non-significant results, in other words, data that was not changed significantly from pre- to post-training must be taken off of the abstract, because it causes mismatch with the actual results.
Legend: n.s. (non-significant) – must be treated as “no changes found”
SG (swimming group)
Decreased
Ø Body mass (n.s.), hip circumference (ok), BCM(n.s.), TBW(n.s.), ECW(n.s.), ICW(n.s.), and FM(n.s.)
Increased
Ø waist circumference(n.s.), FFM(n.s.) and MM(n.s.), WHR (missing in the abstract)
NSG (non-swimming group)
Decreased
Ø BCM (ok), TBW(n.s.), FFM(n.s.), MM(ok), ECW, ICW (ok, but inverted in the abstract)
Increased
Ø Body mass (n.s.), waist/hip circumference (n.s.), and FM (n.s.)
Discussion:
Line 200: “…2-test session (12 200 th week) was significantly lower…”
Comment: The figure must clearly present this information. I recommend display a graph with sample distribution, to better visualize that information.
Line 202: “…anaerobic changes...”
Comment: The finding about lactic acid by authors should not be used as the aerobic enhancement of swimming activities description, on the other hand I recommend proper literature to describe the aerobic changes provided by swimming.
Line 203-204: “…The training protocol also resulted in a change in the deposition of fat in the SG group participants (P<0.01)…”
Comment: Lines 154-156 shows opposite data/description. Review that information.
Overall discussion comment: Non-significant results should not be treated as “slightly” difference but “no difference”, and should not be discussed. The main points to discuss on this work is regarding other physical approaches in similar sample/training and data. Discuss about sedentarism and data collected, muscle mass, hip circumference, ECW, ICW, BCM, focusing on physical activity/wellbeing, health and sample, translating to a scenario of comorbidities e.g. Authors should provide references more often to every given clinical/physiological information.
Limitations:
Comment: Authors recognize in part the major limitations of the study despite I think most part of limitations is on the objectives-methodology match. It’s an interesting work, with a good idea but focused in provides a broadly data about physical changes on women’ body composition, not about creating a method or comparing groups. This work would be better presented if the authors focus were in display body physical composition changes of swimming practice in women, with nutritional detailed information, type of exercise, an intermediated data collection, any data image collection (MRI?). What would be more significant in clinical conditions, such as metabolic syndrome for example.
Conclusion:
Overall comment: Authors should not describe non-significant results as “changes”. ICW and ECW was both reduced, so there wasn’t a “toward movement from intracellular to extracellular”. Reformulate the conclusion focusing with the new findings.
Author Response
Reviewer 1 – answers
Thank you very much for so thorough review of our article. Thank you very much for all the comments that are very valuable to us and have been very helpful in improving our work.
Answers:
Abstract:
Statement: “…This is very important especially for women, for whom the maintenance of lean body in good shape is sometimes a primary consideration…”
“…However, in most cases, this activity is taken randomly and do not produce the desired effects such as reducing body fat…”
Comment: Personal not referenced information
Answer: Thank you very much for that comment but this sentence was formulated based on the analysis of data from this study and available literature. It is not a personal comment. Due to the requirements for the form of this section, I could not put any reference.
Results from Abstract doesn’t match to the displayed in the results section.
Answer: It has been improved.
Introduction:
Line 34: “…Low physical activity of humans leads to changes in the proportions of the basic components of 35 the body. It is well known that this refers mainly to increase in body fat…”
Suggestion: Clarify the phase and insert the proper reference to that information
Answer: It has been inserted.
Line 35: “…Numerous studies have 36 shown that excess body fat and its specific depositing, depending on gender, significantly increases the risk of ischemic heart disease, diabetes, hypertension, or various forms of cancer [6]…”
Suggestion: Don’t use a hole book to reference information so specific like body fat accumulation (obesity) and its risk of certain diseases or cancer. I’d suggest looking for WHO, or AHA, FDA, a most proper source of information.
Answer: It has been improved and proper sources were inserted.
Line 37-40: “…The effects of imbalance in the body of individual body components affect people of all ages. Preservation of the content of body components from an early age ensures the maintenance of good health later in life, when the metabolic processes are slowed down..”
Suggestion: Reference that information.
Answer: It has been inserted.
Line 41-52: broad content not referenced
Suggestion: Do a proper referenced introduction, avoiding hole book, using chapters or original scientific papers, research data, health organs, systematic reviews, etc.
Answer: Thank you so much for such a thorough analysis of the literature on the subject. Proper sources were inserted
Line 50: “…by women participating in various activities...”
Comment: The statement gives the author an impression that the importance of the work it’s not related only to the human health or enhance the global health, but the women health. However, If the authors were actual focusing on this, thus I suggest introducing the manuscript pointing out female aspects that justify this approach (e.g. swimming) in women system and not obesity or because of the fat tissue accumulation. I believe that the fact of a female individuals has been elected for the study not necessarily indicates the hole work has to be constructed on the gender, it’s a sample characteristic.
Answer: Thank you very much for that comment. Because women are the subject of the study, I focused on characterizing this part of the women population undertaking physical activity. The results presented in this paper are part of a large scientific project, where the effectiveness of physical activity currently commonly undertaken by women to supplement knowledge on the optimization of training programs was assessed.
Material and methods:
Line 56: (questions of the study)
“… 1) Will the 12-week swimming training cause changes in body composition of the tested women? and 2) What is the direction and significance of changes caused by the 12-week swimming training?...“
Suggestion: Correct that information to the evaluated, e.g. the primary object is in comparison to sedentary women, and so on. Because, starting from that informed, the control group would be no longer necessary, only a pre-post evaluation.
Answer: thank you very much for that comment. In our opinion, even If I did not use this information, the experiment, which was planned in this way requires the inclusion of a control group, i.e. the group without intervention (training program). In our case, having regard to limitations, inactivity has shown a detrimental effect of sedentary lifestyle. Other author used the same protocol, e.g. Neiva, et al. (2018). The effect of 12 weeks of water-aerobics on health status and physical fitness: An ecological approach. PLoS ONE 13(5), e0198319 (it was recommended by Reviewer 2).
Line 66: “…Additionally, the subjects were asked to analyze their menstrual cycles by measuring vaginal temperature before getting out of bed in the morning…”
Comment: Why? Authors must clarify the reasons and give proper reference to that procedure.
Answer: I placed proper reference.
Line 75: Table 1:
Ø OBS1: on title, I suggest change “…rest value (1) and after the experiment (2)…” to “rest values pre (1) and post (2) 12-week swimming training”
Answer: It was done.
Ø OBS2: BMI SG: 21.9+2.2 is the same value to both moments and there was a decrease of 0.1%, I think that might be some mistake.
Answer: I am very sorry for that mistake. It has been improved.
Line 91: Before and after all 12 weeks experiments or after every experiment? I suggest make this clear. I believe these data were collect twice, one on the exact beginning of the protocol and then 12-weeks later for the same investigator.
Answer: I improved that sentence.
Line 100: about the intensity of the training: it was measured from Karvonen method? Taken everyday or based on the data of the initial test?
Answer: The Karvonen method has not been used as a daily measurement of the incidence of cardiac contractions to determine the rest value of HR because it was very difficult to perform for technical reasons – the training sessions begin at 6.15 a.m. That is why we decided to apply a training intensity assessment based on %HRmax, and Borg’s scale in the case of disturbance or lack of HR recording.
Figure 2: unnecessary.
Answer: Yes, we agree with you. It has been removed.
Results:
Line 128: “…Table 1 shows the anthropometric characteristics of the women from both groups..”
Comment: phase can be removed or place the table 1 on results section.
Answer: I placed Table 1 in section Results
Line 131-133: “…It is noteworthy that at the beginning of the experiment the SG group had more women, but the principle was adopted that eventually would form the SG group of these women (n=17) who participated in at least 95% of all 36 training sessions...”
Comment: The phase is confusing. What do you mean with “more women”? Isn’t 17 per group since the very beginning? Statement should be corrected with the relevant information.
Answer: We wanted to give an information that only 17 (from 24 at the beginning) women completed at least 95% and their data were taken to analyse. The sentence was rewritten.
Line 133: “…at least 95% of all 36 training sessions…”
Comment: what was the participation rate of the swimming group?
Answer: 7 women resigned from participation in the study because of different reasons (health, classes, personal reasons).
Line 166-182: (correlations analysis)
Comment: What was the moment evaluated? Final experiment moment? Correlations doesn’t show causality. I think the Persons can give additional information but, in this case, e.g. the BMI and hip and waist circumference, the BMI implies the mass (weight) evaluation and certainly a higher hip and waist circumference is positively correlated to BMI, there is some unnecessary correlations not related to the given approach/training i.e. showing correlation between swan kilometer per lost of WHR or to the enhancement of heart rate response. I would suggest a different Persons evaluation.
Answer: By showing all these correlations we wanted to show the complexity of the relationship between body composition parameters, the way of depositing fat in the woman's body. We agree with the Reviewer that showing correlation between swan kilometer per loss of WHR or to the enhancement of heart rate response would not make sense.
Line 183-186: Comparison between groups
Comment: There was no intervention on the control group, therefore, there is no reasoning question to compare CT to swimming. If the study was based on two different physical approaches (e.g. on water and off water, cooled vs heated swimming pool) so that would be a reason for this, but in this case there was not. The CT group in this study shows how 12-weeks of sedentary life-style can bring negative significant changes to the women’ body composition.
Answer: This comment is very useful and thank you very much for it. In planning this experiment, we assumed that a group with an intervention (swimming training) and a group without intervention (no swimming training) would be considered. In another experiment within the same project, the intervention was 12-week aerobics training and 12-week Nordic walking training. In this way, we wanted to demonstrate the beneficial impact of such physical activity by comparing it with people with sedentary lifestyles. This experiment is a starting experiment for the next steps in our scientific project, where we will carry out the same protocol for other groups: overweight, obesity and other conditions as you mentioned, e.g. on water and off water, cooled vs heated swimming pool, different structure of single session.
Results displayed on manuscript and the mistakes in comparison to the abstract
All non-significant results, in other words, data that was not changed significantly from pre- to post-training must be taken off of the abstract, because it causes mismatch with the actual results.
Legend: n.s. (non-significant) – must be treated as “no changes found”
SG (swimming group)
Decreased
Ø Body mass (n.s.), hip circumference (ok), BCM(n.s.), TBW(n.s.), ECW(n.s.), ICW(n.s.), and FM(n.s.)
Increased
Ø waist circumference(n.s.), FFM(n.s.) and MM(n.s.), WHR (missing in the abstract)
NSG (non-swimming group)
Decreased
Ø BCM (ok), TBW(n.s.), FFM(n.s.), MM(ok), ECW, ICW (ok, but inverted in the abstract)
Increased
Ø Body mass (n.s.), waist/hip circumference (n.s.), and FM (n.s.)
Answer: It has been improved
Discussion:
Line 200: “…2-test session (12 200 th week) was significantly lower…”
Comment: The figure must clearly present this information. I recommend display a graph with sample distribution, to better visualize that information.
Answer: It was improved.
Line 202: “…anaerobic changes...”
Comment: The finding about lactic acid by authors should not be used as the aerobic enhancement of swimming activities description, on the other hand I recommend proper literature to describe the aerobic changes provided by swimming.
Answer: The other authors used analysis of blood lactate concentration to assess the aerobic enhancement of swimming activity as well, e.g. Wakayoshi, K., Yoshida, T., Ikuta, Y., Mutoh, Y., & Miyashita, M. (1993). Adaptations to six months of aerobic swim training. International Journal of Sports Medicine, 14(07), 368-372.
Line 203-204: “…The training protocol also resulted in a change in the deposition of fat in the SG group participants (P<0.01)….
Comment: Lines 154-156 shows opposite data/description. Review that information.
Answer: The sentence has been rewritten. I am saying about deposition of fat and not about fat mass in this sentence
Overall discussion comment: Non-significant results should not be treated as “slightly” difference but “no difference”, and should not be discussed. The main points to discuss on this work is regarding other physical approaches in similar sample/training and data. Discuss about sedentarism and data collected, muscle mass, hip circumference, ECW, ICW, BCM, focusing on physical activity/wellbeing, health and sample, translating to a scenario of comorbidities e.g. Authors should provide references more often to every given clinical/physiological information.
Answer: Everything was improved according to the comments above.
Limitations:
Comment: Authors recognize in part the major limitations of the study despite I think most part of limitations is on the objectives-methodology match. It’s an interesting work, with a good idea but focused in provides a broadly data about physical changes on women’ body composition, not about creating a method or comparing groups. This work would be better presented if the authors focus were in display body physical composition changes of swimming practice in women, with nutritional detailed information, type of exercise, an intermediated data collection, any data image collection (MRI?). What would be more significant in clinical conditions, such as metabolic syndrome for example.
Answer: I am very thankful for these comments and of course I respect them. The discussion section was supplemented with a deeper analysis of work limitations and their impact on the analysis of results. Your recommendations are very helpful and certainly I will take under consideration in the next step of my scientific project in other group of women.
Conclusion:
Overall comment: Authors should not describe non-significant results as “changes”. ICW and ECW was both reduced, so there wasn’t a “toward movement from intracellular to extracellular”. Reformulate the conclusion focusing with the new findings.
Answer: It has been rewritten.

Round 2
Reviewer 3 Report
I must say your manuscript has been significantly improved (P